# Automated Denudation of Oocytes

**DOI:** 10.3390/mi13081301

**Published:** 2022-08-12

**Authors:** Rongan Zhai, Guanqiao Shan, Changsheng Dai, Miao Hao, Junhui Zhu, Changhai Ru, Yu Sun

**Affiliations:** 1School of Mechatronic Engineering and Automation, Shanghai University, Shanghai 200444, China; 2Department of Mechanical and Industrial Engineering, University of Toronto, Toronto, ON M5S 3G8, Canada; 3School of Mechanical and Electrical Engineering, Research Center of Robotics and Micro Systems, Soochow University, Suzhou 215021, China; 4School of Electronic and Information Engineering, Suzhou University of Science and Technology, Suzhou 215009, China

**Keywords:** ICSI, oocyte denudation, robotic system

## Abstract

Denudation is a technique for removal of the cumulus cell mass from oocytes in clinical intracytoplasmic sperm injection (ICSI). Manual oocyte denudation requires long training hours and stringent skills, but still suffers from low yield rate and denudation efficiency due to human fatigue and skill variations across operators. To address these limitations, this paper reports a robotic system for automated oocyte denudation. In this system, several key techniques are proposed, including a vision-based contact detection method for measuring the relative z position between the micropipette tip and the dish substrate, recognition of oocytes and the surrounding cumulus cells, automated calibration algorithm for eliminating the misalignment angle, and automated control of the flow rate based on the model of oocyte dynamics during micropipette aspiration and deposition. Experiments on mouse oocytes demonstrated that the robotic denudation system achieved a high yield rate of 97.0 ± 2.8% and denudation efficiency of 95.0 ± 0.8%. Additionally, oocytes denuded by the robotic system showed comparable fertilization rate and developmental competence compared with manual denudation. Our robotic denudation system represents one step towards the automation and standardization of ICSI procedures.

## 1. Introduction

Human infertility has remained a highly prevalent global condition. Infertility is estimated to affect between 8% and 12% of reproductive-aged couples worldwide [1]. Intracytoplasmic sperm injection (ICSI) is a powerful therapeutic tool for the treatment of couples with severe infertility, and thousands of infertile couples have benefited from its use [2,3,4]. In ICSI procedures, the cumulus–oocyte complexes (COCs) are obtained through follicular aspiration, and the cumulus cells surrounding the oocyte must be removed to facilitate the injection of sperm [5] and the evaluation of oocyte morphology, in particular, the nuclear maturation stage for the extrusion of the first polar [6,7,8]. The removal of the cumulus cells from the oocyte is called oocyte denudation [9].

At present, oocyte denudation is done manually in ICSI clinics globally. An operator looks through the microscope eyepieces and manipulates oocytes using a mouth-controlled or hand-controlled pipette. Oocyte denudation is performed in two steps [10,11,12]. First, COCs are enzymolyzed in hyaluronidase to weaken the bonds between cumulus cells [13]. Due to the potential toxicity of the hyaluronidase on the oocytes, the enzymatic time (~40 s) is critical, which, however, is difficult to strictly control in manual operation. Then, cumulus cells are removed from oocytes by the pipette aspiration and deposition in a series of mediums without an enzyme. During pipette aspiration and deposition, it is difficult to recognize and manipulate the oocytes and the surrounding cumulus cells, especially when the number of the oocytes is large, because of their small sizes. In addition, manual denudation is a laborious and demanding task, and due to poor reproducibility and inconsistency across operators, the yield rate and denudation efficiency also vary significantly.

In the past few decades, robotic techniques have demonstrated great progress in assisting and standardizing clinical procedures. Lu et al. developed the first robotic ICSI system capable of recognition and manipulation of sperm and oocytes for automated ICSI procedures [14]. Leung et al. integrated the vision-based contact detection method to determine the relative vertical position between the micropipette tip and surface of the Petri dish, and the XY stage motion control algorithms to move the sperm of interest to the center of view for robotic immobilization [15]. Zhang and Lu developed robotic manipulation techniques for automated aspiration of a single sperm cell into a micropipette and accurate positioning of the sperm to a target position based on the model of sperm dynamics inside the micropipette [16]. Liu et al. developed a robotic vitrification system [17] that was able to pick and place an embryo at a time. This system was embedded with two contact detection methods to determine the relative z positions of the vitrification micropipette, embryos, and vitrification straw. In this work, Zhang and Liu [18] further built the model of embryo dynamics to aspirate each embryo with minimum volume of excess medium.

Efforts have been made to automate the denudation process. Zeringue et al. designed a microfluidic channel which pushed the cumulus to the sides of the oocyte and removed them at right corners [19,20]. However, this device required manual control and switch of multiple fluid flows. Weng et al. developed a microfluidic device with a series of jagged-surface constriction microchannels to denude oocytes from the surrounding cumulus cell mass in a continuous fluid flow [21]. However, the denudation process is complex and time-consuming. Mokhtare et al. recently introduced a non-contact oscillatory microfluidic channel that used a bas-relief structure to twist the flow inside the channel [22,23]. The additional preparation and delicate loading of the COCs to the device made the overall process prohibitively cumbersome and time-consuming. Reeder and Monson [24] denuded embryos from cumulus cell mass by high-speed vortexing. However, embryos were easy to be damaged due to strong forces applied to them. No attempt has been made to automate denudation using a robotic approach.

In this study, using mouse models, we developed a robotic system that was able to denude oocytes from the surrounding cumulus cell mass. The system achieved oocyte denudation with the following techniques: (1) utilization of a contact detection method to determine the relative z position of the micropipette tip and the dish substrate; (2) automated recognition of oocytes and the surrounding cumulus cells; (3) algorithms for eliminating the misalignment angle of XY stage; and (4) automated control of the flow rate from a syringe pump. The experimental results demonstrated that the yield rate and denudation efficiency achieved by the robotic system were higher than those of the manual denudation, and no statistically significant differences in the fertilization rate and the developmental potential were found. Furthermore, our robotic denudation system has the translational potential to process human samples in an automated and standardized manner using clinical setup.

## 2. System Overview

### 2.1. System Setup

As shown in Figure 1a, the system is built on a standard upright microscope (Olympus SZX16) that is equipped with motorized focusing (Z). Mounted on the microscope is an XY stage (MLS-1740, Micro-Nano Automation Institute Co., Ltd., Suzhou City, China.), which has a travel range of 60 mm and a resolution of 1 μm along both axes. A heating stage (MATS-UAXKP-D, Tokai Hit Co., Ltd., Fujimiya City, Japan.) is placed on the XY stage to maintain the samples in a custom-designed multiwell dish at 37 °C. The multiwell dish consists of four wells with 2 mm depth and the space between each well is equal, where oocytes are thoroughly immersed, and meet the requirements of automatic denudation and positioning (see Figure 1b). A micropipette (MDP-125-0, ORIGIO Inc., Delaware, USA.) with 125 μm tip diameter is mounted on a robotic micromanipulator (MX7600R, Siskiyou, Inc., Grants Pass, OR, USA.). A 100 µL glass syringe is mounted on a motorized pump (PHD 22/2000, Harvard Apparatus Company, Holliston, MA, USA.) to control the microscope for oocyte aspiration and deposition. A camera (scA1300-32 gm, Basler, Inc., Lubeck, Germany) is connected to the microscope to obtain visual feedback. Figure 1c summarizes the overall control architecture of the system. Visual feedback of the positions of the micropipette tip, oocytes, and the surrounding cumulus cells is used to form an image-based visual servo control system. The robotic micromanipulator, XY stage, focusing (Z) motor, and the motorized syringe pump are cooperatively controlled for the pick-and-place of oocytes.

### 2.2. System Operation

C57BL/6 female mice (12–14 weeks old, Genepharma Laboratories, Suzhou, China) were injected intraperitoneally with 5 IU of pregnant mare serum gonadotrophin (PMSG), followed by a second injection of 5 IU of human chorionic gonadotropin (HCG) 48 h later. COCs were retrieved from the ampulla of each oviduct 15 to 16 h after the second injection, and placed in hyaluronidase. First, the system lowers the micropipette to 1 mm vertically below the surface of hyaluronidase in well Q1. A contact detection method [25,26] is then used by the system to determine the relative z position of the micropipette tip and the well (Q1) dish substrate. With both micromanipulator and focusing (Z) motor motion, the position of the micropipette tip is set at 25 μm above the well (Q1) dish substrate and 150 μm left to the center of view. After searching and taking oocytes to the view of the microscope, the system can recognize contours of all oocytes and return the center coordinate of each oocyte contour by image processing. Subsequently, oocytes are chosen by a computer program and brought to the center of view via the XY stage movement, sequentially. The micropipette is then lowered to vertically align with the target oocyte. After all oocytes are aspirated into the micropipette, the micropipette is then lifted to 55 μm above the top surface of the multiwell dish and moved to next well (Q2). Following the same method, every oocyte is automatically brought to the center of view for micropipette aspiration and deposition in the well (Q2). For an oocyte in this process, the cumulus cells surrounding the oocyte outside the micropipette and the oocyte position inside the micropipette are detected by image processing. The detection results are fed back to system for controlling the flow parameters of the syringe pump, which enables complete oocyte denudation without oocyte loss. After complete denudation in well Q3, all oocytes are aspirated into the micropipette and transferred to the next well (Q4). Finally, the micropipette returns to the initial position above the first well (Q1) for the next denudation process. During the denudation process, the oocytes are enzymolyzed in the hyaluronidase for less than 40 s to minimize enzyme toxicity.

## 3. Key Methods

### 3.1. Contact Detection and Micropipette Tip Identification

Robotic cell manipulation is commonly conducted under an optical microscope with limited depth of field, making position detection along the *Z*-axis difficult. The contact detection method reported by our previous research [25,26] accurately determines the relative *Z*-axis position of the micropipette tip and the well dish substrate. In the contact detection process, a 230 × 200-pixel region of interest (ROI) is built at the micropipette tip to alleviate computation complexity and allow real-time performance. As shown in Figure 2, the micropipette tip is detected by finding the rightmost end of micropipette wall edges. The algorithm can detect the micropipette tip from the unchanged background for each frame image by a series of image processing.

### 3.2. Oocyte and Cumulus Cell Recognition

After COCs are enzymolyzed in the hyaluronidase, there are still some cumulus cells surrounding the oocyte to be removed by repeated aspiration and deposition. In this process, a recognition algorithm (see Figure 3) is developed to recognize oocyte and cumulus cells as system feedback, which ensures that the oocyte is denuded completely. As the image intensities of the oocyte surface parts vary greatly, an oocyte contour is incorrectly recognized as many small contours by traditional recognition algorithms. To improve the processing speed and the contour recognition precision, the filling algorithm is used to fill the holes caused by varied image intensity of oocyte surface parts (Figure 3d). The masses of the oocyte and cumulus cells are obtained computing their volume.

### 3.3. Calibration with XY Stage

When the CCD camera is manually installed in the microscope, there will be an installation angle of the CCD camera, which causes misalignment between the stage coordinate system and the image coordinate system. Due to the misalignment angle, oocytes are not positioned by the XY stage at the center of view accurately, resulting in poor performance of oocyte aspiration, especially when oocytes are far from the center of view. Manual calibration involves repeated rotation of the CCD camera, which is time-consuming and imprecise. In this system, an automated calibration algorithm is developed.

As shown in Figure 4, the image coordinates of point A and point B are (*X*_0_,*Y*_0_) and (*X*_1_,*Y*_1_), and the stage coordinates of point A and point B are (*X*_0_′,*Y*_0_′) and (*X*_1_′,*Y*_1_′). Due to the influence of the installation angle, the position of point A in the image is moved to the position of point B in the image accurately by sending instruction of stage displacement (*X*_1_′ − *X*_0_′, *Y*_1_′ − *Y*_0_′) instead of known image displacement (*X*_1_ − *X*_0_, *Y*_1_ − *Y*_0_) to the stage.
(1)$$\left[\begin{array}{c} X_{1}\prime -X_{0}\prime \\ Y_{1}\prime -Y_{0}\prime \\ 1 \end{array}\right]=\left[\begin{array}{ccc} \cos(\theta ) & -\sin(\theta ) & 0\\ \sin(\theta ) & \cos(\theta ) & 0\\ 0 & 0 & 1 \end{array}\right]\left[\begin{array}{c} X_{1}-X_{0}\\ Y_{1}-Y_{0}\\ 1 \end{array}\right]$$
where *θ* is the misalignment angle between the stage coordinate system and the image coordinate system.

For the automated calculation of the stage displacement, the misalignment angle needs to be obtained. As shown in Figure 5a, two adjacent frames in the same stage stepping direction with partially overlapping pictures are selected and processed by a computer for image mosaic construction.

In the process, the SIFT algorithm [27] is used for image feature extraction, the RANSAC algorithm [28] is applied to image feature matching to eliminate outliers, and key matching points are put into use for image mosaic construction. Finally, the stage stepping errors *dy* (Pix) and *dx* (Pix) are calculated through the resulting mosaic diagram as shown in Figure 5b. Then, the misalignment angle *θ* is obtained by
(2)$$\theta =\arctan\frac{dy}{dx}$$

After calculation of the misalignment angle, the instruction of stage displacement (*X*_1_′ − *X*_0_′, *Y*_1_′ − *Y*_0_′) is sent to the stage, which ensures that the position of point A in the image is moved to the position of point B in the image accurately for denuding oocytes. Therefore, the XY stage is calibrated automatically each time instead of manual calibration.

### 3.4. Fluid Flow Control

After being enzymolyzed in the hyaluronidase under controlled timing, the oocytes have different masses of residual cumulus cells surrounding them. By manual control of flow rate, not all oocytes are denuded completely by the operator. Moreover, if the induced fluid shear force is larger than the physiological shear force of the oocyte, the oocyte will be damaged. The robotic system can control the flow rate using the motorized syringe pump to remove the residual cumulus cells and protect oocytes by modeling oocyte dynamics during micropipette aspiration and deposition.

Once the oocyte enters the micropipette, its motion is confined by the micropipette wall and moves together with the medium without relative motion. Therefore, relative motion only exists outside the micropipette caused by the aspiration and deposition flow.

As shown in Figure 6, the drag force (*F_d_*) generated by micropipette aspiration flow [17] is
(3)$$F_{d}=\frac{1}{2A}\rho Q^{2}C_{d}$$
*A* is the cross-sectional area of the micropipette, *ρ* is the M2 medium density, *Q* is the flow rate controlled by the motorized syringe, and *C_d_* is the drag coefficient.

To calculate the fluid shear force (*F_s_*), according to Newton’s law, the dynamic equation of the oocyte is
(4)$$(M+m)\overset{‥}{x}+F_{f}+F_{s}=F_{d}$$
*M* is the mass of the oocyte, *m* is the mass of cumulus cells surrounding the oocyte, *x* is the displacement of the oocyte, and *F_f_* is the friction between the oocyte and cumulus cells.

As shown in Figure 3f, according to the equation for a sphere’s volume, the equation of mass is
(5)$$\left(M+m)=\frac{4}{3}\pi \rho_{0}(R^{3}+r^{3}\right)$$

*ρ*_0_ is the cell density, *R* is the radius of the oocyte, and *r* is the radii of cumulus cells.

The motion acceleration is constant in the short time of micropipette aspiration. Based on the acceleration law, the equation of acceleration is
(6)$$\overset{‥}{x}=\frac{v^{2}}{2s}=\frac{1}{2s}\frac{Q^{2}}{A^{2}}$$
*v* is the speed of the oocyte at the micropipette tip, and *s* is the initial displacement from the position of the oocyte to the position of the micropipette tip.

According to the friction law, the equation of friction is
(7)$$F_{f}=(M+m)g\mu$$
*μ* is the friction coefficient.

Substituting (5)–(7) into (4), we can obtain
(8)$$\frac{2}{3}\frac{\pi \rho_{0}Q^{2}}{A^{2}s}(R^{3}+r^{3})+\frac{4}{3}\pi \rho_{0}g\mu (R^{3}+r^{3})+F_{s}=\frac{1}{2A}\rho Q^{2}C_{d}$$
(9)$$F_{s}=\frac{1}{2A}\rho C_{d}Q^{2}-\frac{2}{3}\pi \rho_{0}(\frac{Q^{2}}{A^{2}s}+2g\mu )(R^{3}+r^{3})$$

To remove the cumulus cells from the oocyte, the fluid shear force (*F_s_*) must be larger than the adhesion force between the cumulus cells and the oocyte, but smaller than the physiological shear force (*F_mx_*) of the oocyte, to prevent damage.
(10)$$mgu_{s}\leq F_{s}\leq F_{mx}$$
(11)$$\sqrt{\frac{8\pi A^{2}s\rho_{0}g(\mu R^{3}+\mu r^{3}+\mu_{s}r^{3})}{3\rho C_{d}As-4\pi \rho_{0}(R^{3}+r^{3})}}\leq Q\leq \sqrt{\frac{2A^{2}s(4\pi \rho_{0}g\mu R^{3}+4\pi \rho_{0}g\mu r^{3}+3F_{mx})}{3\rho C_{d}As-4\pi \rho_{0}(R^{3}+r^{3})}}$$
*μ_s_* is the adhesion coefficient. Meanwhile, the oocyte dynamics outside the micropipette during deposition is the same as the dynamics during aspiration.

During the micropipette aspiration and deposition, the technique for oocyte and cumulus cell recognition is used to compute the radii of oocytes and cumulus cells, and provide flow rate feedback for syringe pump control, forming an image-based visual servo control system. The robotic system is capable of denuding oocytes completely and protecting oocytes from damage.

In the manual denudation process, the oocytes tend to move deeply into the micropipette and disappear during aspiration, due to the uncertainty of the oocyte position inside the micropipette. In contrast, in the robotic system, the oocyte position inside the micropipette is detected in real time by image processing (see Figure 7). If the oocyte position is beyond the left limit position of the micropipette, the robotic system can stop the flow rate of the syringe pump immediately and avoid oocyte loss.

## 4. Results and Discussion

### 4.1. Visual Recognition

The recognition success rates of the oocytes and the surrounding cumulus cells were defined to quantify the robotic system’s performance of visual recognition. Recognition was considered successful when the algorithm focused on each oocyte or the surrounding cumulus cells, and indicated the XY position and radius of them. Six visual recognition experiments of 100 mouse oocytes were conducted. For oocyte recognition in the multiwell dish, 100 out of the 100 oocytes were successfully recognized. Although some cumulus cells removed from oocyte were dispersed in the multiwell dish, none of them were falsely detected as oocytes. However, for the surrounding cumulus cell recognition in the multiwell dish, five oocytes’ cumulus cells were not recognized, yielding a success rate of 95%. In the failure cases, due to the lower contrast between the cumulus cells and the background, the contour edges of the cumulus cells were not recognized correctly. To overcome this problem, contrast-enhancing imaging techniques, such as phase-contrast imaging and differential interference contrast (DIC) imaging, can be used to increase the contrast of cumulus cell edges for further improvement of cumulus cell detection accuracy. Additionally, the speed of the recognition algorithm achieved real-time performance at 30 Hz.

### 4.2. Automated Calibration

In the process of denudation, oocytes need to be moved to the center of view for micropipette aspiration. Due to the influence of the misalignment angle *θ*, the oocytes were not accurately positioned at the center after the XY stage movement, which caused poor performance of oocyte denudation. In order to further measure the accuracy of automated calibration, the calibration error test experiment was conducted. A total of 50 mouse oocytes were selected randomly and moved to the center via the XY stage movement (see Figure 8) by automated calibration and manual calibration (rotating the CCD camera manually).

The experimental results are described in Figure 9. The number of oocytes with lower than 5 μm in X error and Y error for the manual group was 0. However, the number of oocytes with lower than 5 μm in X error and Y error for the automated group was 50, which means that the automated group had lower calibration error than the manual group. In the manual group, there were nine oocytes near the edge of the field of view. Due to the uncertainty of the manual calibration, the misalignment angle *θ* remained large, resulting in large X error and Y error of oocyte positioning. As a result, after moving the oocytes to the center of view via the XY stage movement, four oocytes were outside the field of view, and five oocytes were beyond the working range of micropipette aspiration. Therefore, these nine oocytes with large XY positioning error failed to be aspirated into the micropipette, and were not denuded, leading to a lower rate of oocytes denuded in the manual group. Due to the mechanical displacement error of the stage caused by mechanical wear, the calibration error was not eliminated completely by the automated calibration method, but it was within the acceptable range for micropipette aspiration.

### 4.3. Yield Rate and Denudation Efficiency

The yield rate and denudation efficiency of oocytes were measured to further quantify the performance of the robotic system. The yield rate was defined as the ratio between the number of denuded oocytes and the total number of oocytes. The denudation efficiency was defined as the ratio between the number of completely denuded oocytes and the total number of oocytes. Five experiments of 187 mouse oocytes were conducted using robotic denudation and manual denudation.

As summarized in Table 1, the robotic system achieved a significantly higher yield rate than the manual operation (97.0 ± 2.8% versus 86.9 ± 1.7%). Manual denudation was performed by a skilled operator using a mouth-controlled pipette, and a few oocytes were lost inside the micropipette during the process of aspiration, due to the manual variations in flow rate control and the uncertainty of the oocyte position inside the micropipette. On the contrary, robotic denudation detected the oocyte position inside the micropipette in real time and accurately controlled the flow rate, accordingly. The ability of the robotic system to effectively calibrate XY stage also enabled the robotic system to achieve a higher yield rate. The robotic group produced a significantly higher denudation efficiency than the manual group (95.0 ± 0.8% versus 81.2 ± 2.7%). The higher denudation efficiency produced by robotic system was attributed to the fluid shear force control based on the oocyte dynamics modeling and the technique for oocyte and cumulus cell recognition.

### 4.4. Fertilization and Developmental Potential

In ICSI procedures, the fertilization rate and development potential of oocytes are two important indexes concerned by embryologists. In order to ensure that the robotic denudation process does not damage the fertilization and development potential of the denuded oocytes, we compared ICSI procedures using a total of 206 mouse oocytes denuded by our robotic system and manual operation. The fertilization rate was defined as the ratio between the number of two-cell embryos developed and the number of oocytes denuded. The blastocyst rate was defined as the ratio between the number of blastocysts developed and the number of two-cell embryos. As shown in Figure 10a,b, the mean fertilization rate and blastocyst rate of oocytes denuded by the robotic system for ICSI were 97.8 ± 2.2% and 79.8 ± 1.2%, significantly higher when compared with the rates of 91.5 ± 2.1% and 78.5 ± 1.0% by manual denudation (*p* = 0.19 > 0.05 and *p* = 0.92 > 0.05, Student’s *t* test). Therefore, it is evident that the fertilization and development potential of oocytes denuded by the robotic system were not damaged, compared with the results of manual operation. For each repeat of the ICSI experiments, the robotic and manual denudation procedures were conducted in parallel on the same day. ICSI was then performed using the sperm from the same male mouse. Figure 10c,d displays the representative images of two-cell embryos and blastocysts produced by ICSI.

## 5. Conclusions

This paper presented an automated robotic denudation system capable of removing the cumulus cells from oocytes in clinical ICSI procedures. The system was integrated with a contact detection method to determine the relative z position of the micropipette tip and the well dish substrate. Computer vision algorithms were developed to recognize the position and radius of oocytes and the surrounding cumulus cells, which provided visual feedback for the system to ensure that oocytes were denuded completely. An automated calibration technique was developed to eliminate the misalignment angle and accurately moved the oocyte to the center of view by XY stage movement, which ensured that the oocyte was able to be aspirated into the micropipette for denudation. The flow rate from the syringe pump was controlled to remove the residual cumulus cells and protect oocytes from damage during micropipette aspiration and deposition. These techniques enabled the robotic system to achieve a higher yield rate of 97.0 ± 2.8% and denudation efficiency of 95.0 ± 0.8%, compared to manual denudation. Although not statistically significant in fertilization rate and blastocyst rate compared to manual denudation, our robotic denudation system has the potential to standardize oocyte denudation and improve the performance of ICSI procedures.

## Figures and Tables

**Figure 1 micromachines-13-01301-f001:**
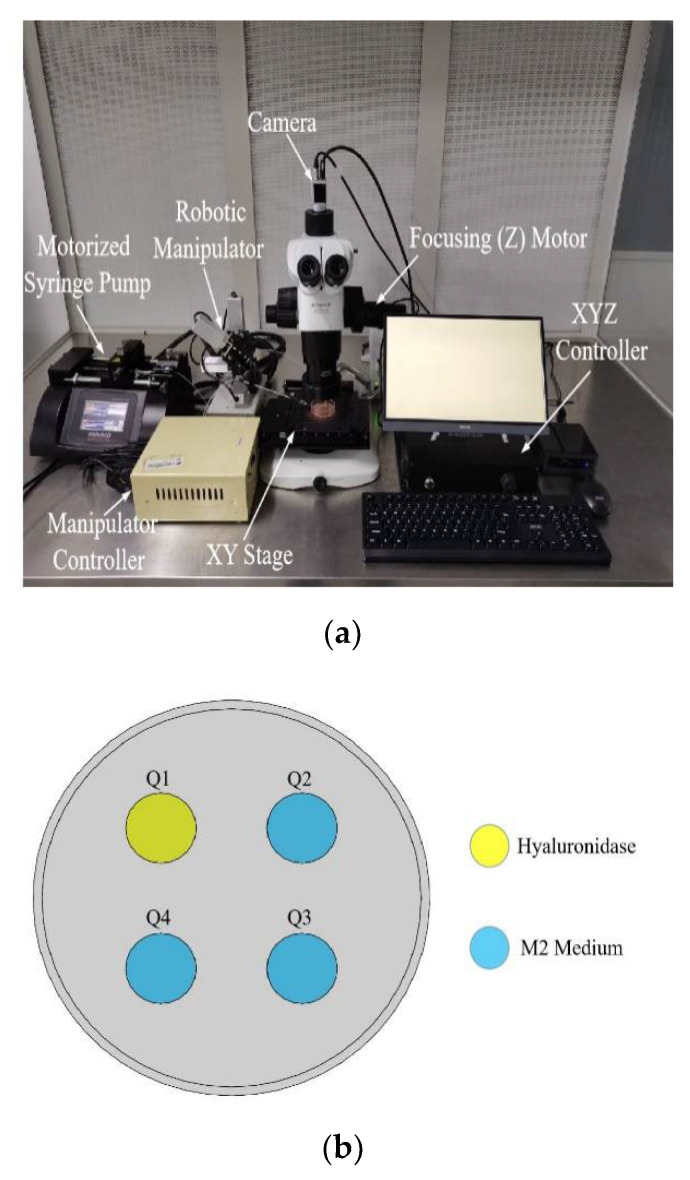
(**a**) Robotic denudation system setup; (**b**) Multiwell dish marked sequentially. Q1 filled with 100 μL hyaluronidase covered with mineral oil, Q2, Q3, Q4 filled with 80 μL M2 mediums covered with mineral oil; (**c**) The system control architecture. E_P_ is the target position of oocytes and cumulus cells. M_P_ is the target position of the micropipette tip for picking up and placing oocytes.

**Figure 2 micromachines-13-01301-f002:**
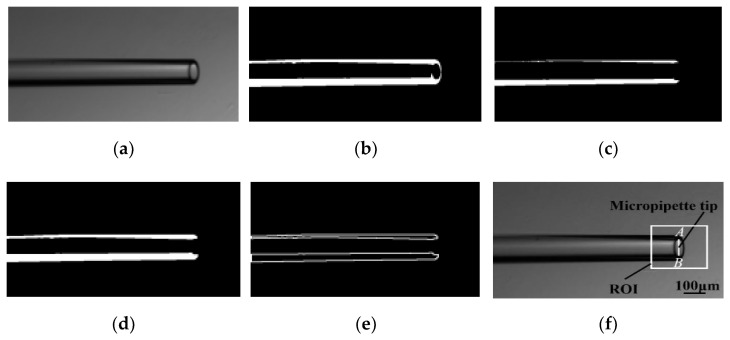
Image sequence of micropipette tip identification. (**a**) Gaussian low-pass filter for denoising image; (**b**) Thresholding method for binarizing low-noise image; (**c**,**d**) Morphological erosion and dilation for eliminating small objects in the binarized image; (**e**) Contour detection for finding the top and bottom edges of micropipette walls; (**f**) Micropipette tip detected at the rightmost end of micropipette wall edges by finding the point A and the point B. Point A is the rightmost point on the top edges of micropipette walls; Point B is the rightmost point on the bottom edges of micropipette walls.

**Figure 3 micromachines-13-01301-f003:**
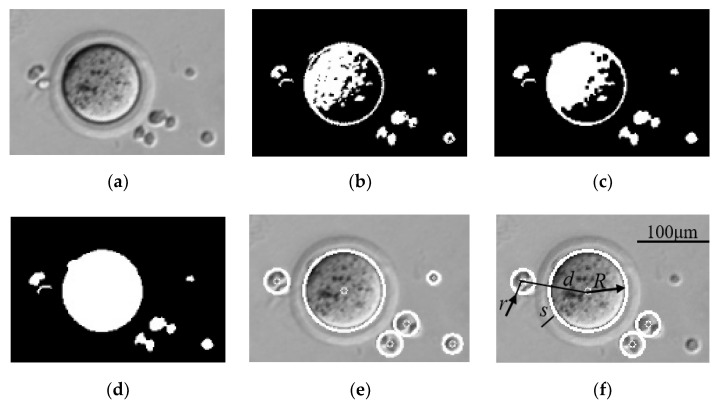
Image sequence of oocyte and cumulus cell recognition. (**a**) Gaussian low-pass filtering image; (**b**) Adaptive thresholding with the Otsu method; (**c**) Morphological close operation creates a smooth oocyte; (**d**) Filling operation fills the holes of oocyte surface; (**e**) Least square fitting to obtain circle contours; (**f**) Oocyte and cumulus cell recognition based on contour area and center distance between oocyte and cumulus cells, where *d* is center distance between oocyte and cumulus cells, *R* is radius of oocyte, *r* is radii of cumulus cells, *s* is thickness of zona pellucida, s ≈ 8 μm.

**Figure 4 micromachines-13-01301-f004:**
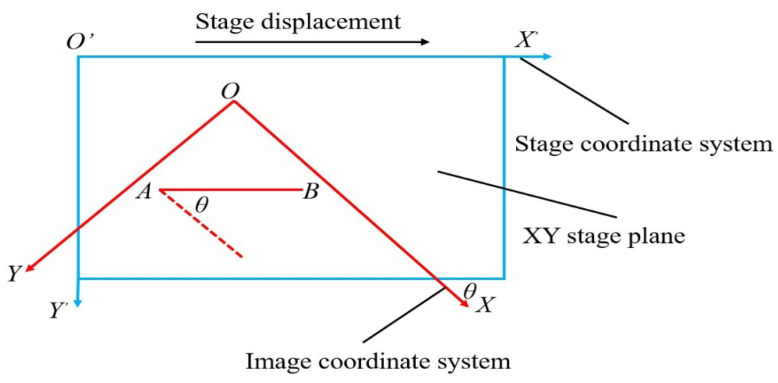
Schematic diagram of coordinate systematic misalignment angle.

**Figure 5 micromachines-13-01301-f005:**
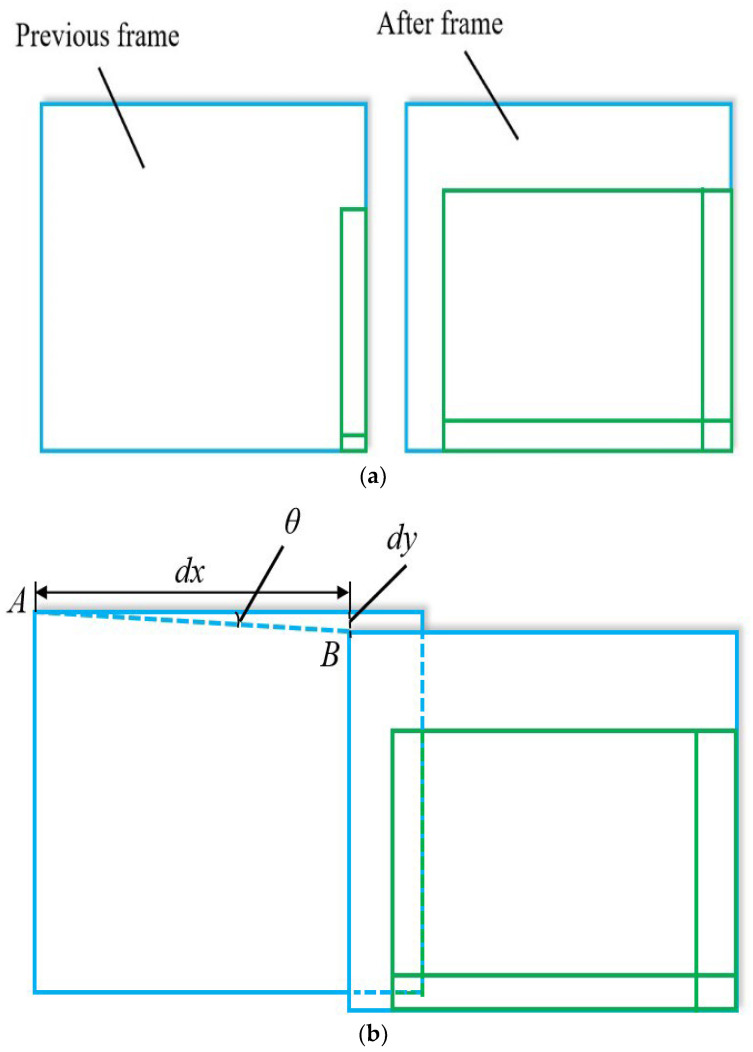
Calculation of the misalignment angle. (**a**) Two adjacent frames; (**b**) Image mosaic.

**Figure 6 micromachines-13-01301-f006:**
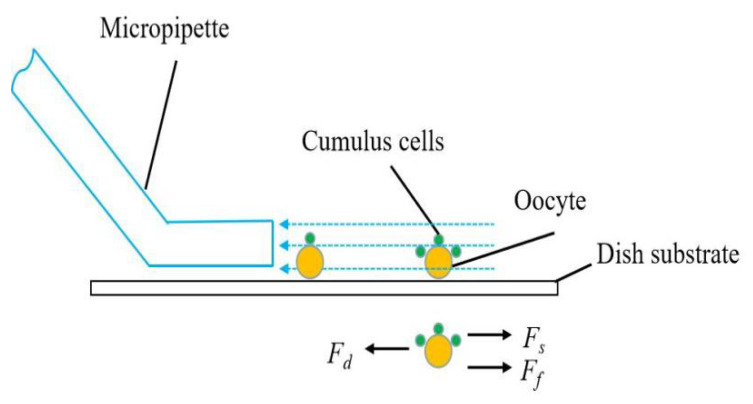
The schematic showing the oocyte dynamics outside the micropipette during aspiration.

**Figure 7 micromachines-13-01301-f007:**
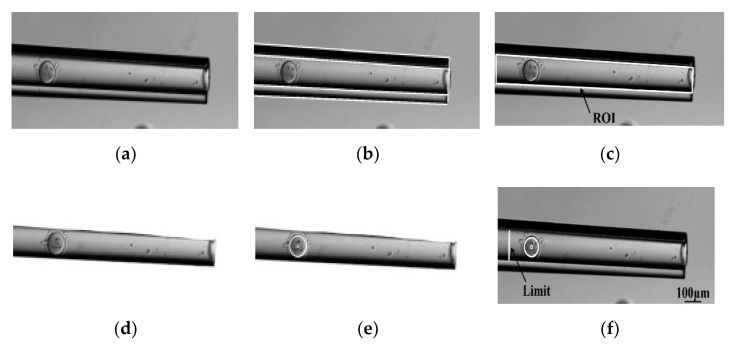
Image sequence of oocyte recognition inside the micropipette. (**a**) Smoothed image; (**b**) Hough transform line detection; (**c**) ROI containing the inside of the micropipette; (**d**) Extracted ROI image; (**e**) Oocyte recognition in (**d**); (**f**) Oocyte recognition within limit.

**Figure 8 micromachines-13-01301-f008:**
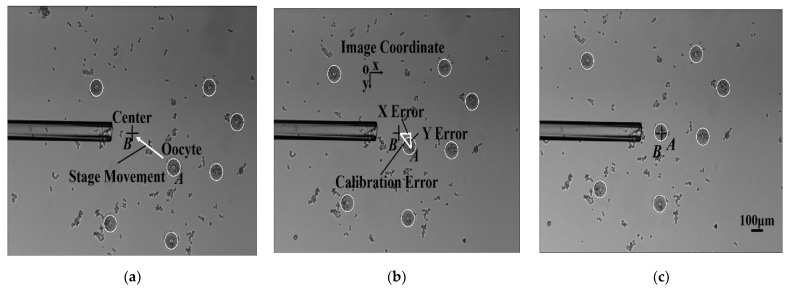
Comparison of two calibration methods. Oocyte A was moved to the center B by stage movement. After stage movement, the calibration error between the oocyte A position and the center B position was calculated. The calibration error was divided into the X error and the Y error along the x and y direction of the image coordinates, respectively. (**a**) Before stage movement; (**b**) After stage movement by manual calibration; (**c**) After stage movement by automated calibration.

**Figure 9 micromachines-13-01301-f009:**
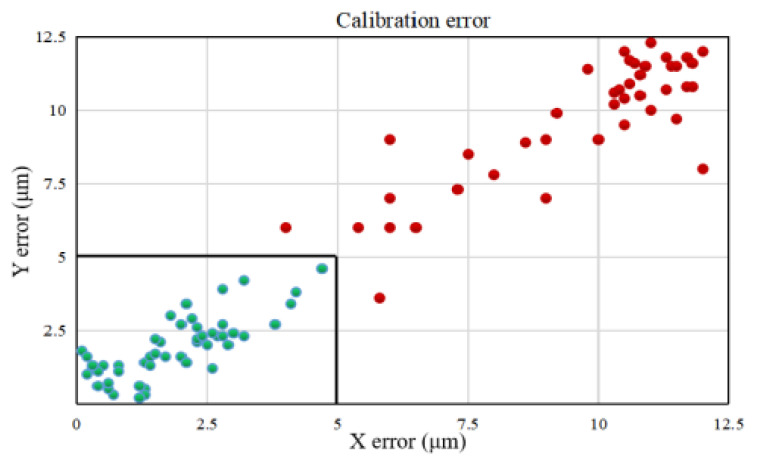
The calibration error of two calibration methods. The green dots are calibration errors of oocytes in the automated group, and the red dots are calibration errors of oocytes in the manual group.

**Figure 10 micromachines-13-01301-f010:**
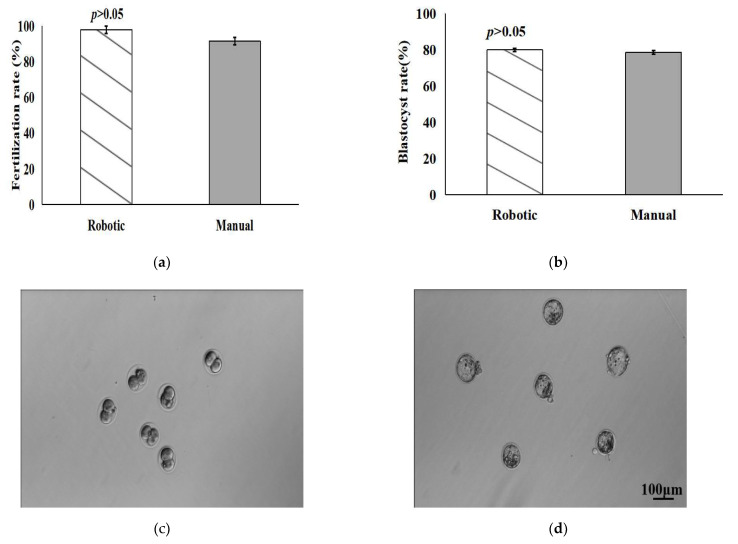
Fertilization and development potentials of oocytes denuded by robotic system and manual operation for ICSI. (**a**,**b**) Fertilization rate of oocytes denuded and blastocyst rate of two-cell embryos; (**c**,**d**) Representative images of two-cell embryos and blastocysts produced by ICSI.

**Table 1 micromachines-13-01301-t001:** Experimental results of mouse oocyte denudation.

Experiment	1	2	3	4	5	Mean ± SD
Manual yield rate (%)	85 (17/20)	87.5 (14/16)	86.7 (13/15)	85.7 (12/14)	89.5 (17/19)	86.9 ± 1.7
Robotic yield rate (%)	100 (25/25)	95.2 (20/21)	100 (18/18)	94.1 (16/17)	95.5 (21/22)	97.0 ± 2.8
Manual denudation efficiency (%)	80 (16/20)	81.3 (13/16)	80 (12/15)	85.7 (12/14)	78.9 (15/19)	81.2 ± 2.7
Robotic denudation efficiency (%)	96 (24/25)	95.2 (20/21)	94.4 (17/18)	94.1 (16/17)	95.5 (21/22)	95.0 ± 0.8

## Data Availability

Not applicable.

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
