# Peer review of "Automated Denudation of Oocytes"

_micromachines, 2022, doi:10.3390/mi13081301_

Round 1

Reviewer 1 Report

The title of this paper is Automated Denudation of Oocytes. In this paper, the author propose a robotic denudation system capable of eliminating limitations in manual operation. The author developed computer vision algorithms to recognize the position and radius of oocyte and its surrounded cumulus cells, which provided visual feedback as the system to ensure that oocyte was denuded completely. Also the author has developed automated calibration technique to eliminate the misalignment angle and make oocyte accurately moved to the center of view by XY stage, which ensured that oocyte was aspired into micropipette and denuded. The flow rate of syringe pump was controlled to remove the residual cumulus cells and protect oocytes from damage during micropipette aspiration and deposition. These techniques enabled the robotic system to achieve a higher yield rate of 97.0±2.8% and denudation efficiency of 95.0±0.8% compared to manual denudation successfully. However, some problems need to be revised as follows:

1.   Please add more information about Equation(5)-(8). There are several  variables such as Ff, M, m which are need to be defined in the sentences.

2 .In the Chapter 4.1, the author explained the failure cases, due to the lower contrast between its surrounded cumulus cells and background. Please add improvement points to overcome your system about this problem in this section.

3.   The author explained that , "in the manual group, 9 oocytes with more X error and in Y error are not aspired into micropipette, and are not denuded, which means that the manual group has lower rate of oocytes denudated than the automated group where all 50 oocytes are denuded". Could it be possible to add more information to justify this phenomenon?

4.  The size of font in the figure 4,7,8,10 is too small and blur. Please improve them.

Author Response

Dear Reviewer,

Thank you very much for the positive and constructive comments.

We have carefully addressed all the comments in the following responses and revised the manuscript accordingly. Please see the attachment.

Reviewer 2 Report

This paper uses microneedle based robotic approach to achieve automated Denudation of Oocytes. The structure of the manuscript is good, and the authors have described the process well. The introduction section is fine, and the other sections of the article are easy to understand. Although the manuscript is well written still the following comments need to be addressed.

1.     There are several errors in writing for example a full stop after references at line number 74 and space after “algorithm” in line 164.

2.     Paper should be sent to a professional English editing service for polishing.

3.     Model numbers of hardware items should also be included in section 2.1

4.     ROI in line number 147 should be defined

5.     It is not clear which method is finally used to detect the end of the micropipette Figure 2 (f).

Author Response

(The authors gave the same response as above.)

Round 2

Reviewer 1 Report

I have confirmed that the paper has been revised. This paper is worth to be published in Micromachine.